# The Impact of Spinopelvic Mobility on Arthroplasty: Implications for Hip and Spine Surgeons

**DOI:** 10.3390/jcm9082569

**Published:** 2020-08-08

**Authors:** Henryk Haffer, Dominik Adl Amini, Carsten Perka, Matthias Pumberger

**Affiliations:** Center for Musculoskeletal Surgery, Charité University Medicine Berlin, Charitéplatz 1 10117 Berlin, Germany; dominik.adl-amini@charite.de (D.A.A.); carsten.perka@charite.de (C.P.); matthias.pumberger@charite.de (M.P.)

**Keywords:** total hip arthroplasty, dislocation, spine surgery, spinopelvic alignment, spondylodesis, spinal fusion, lumbar fusion, spinal balance

## Abstract

Spinopelvic mobility represents the complex interaction of hip, pelvis, and spine. Understanding this interaction is relevant for both arthroplasty and spine surgeons, as a predicted increasing number of patients will suffer from hip and spinal pathologies simultaneously. We conducted a comprehensive literature review, defined the nomenclature, summarized the various classifications of spinopelvic mobility, and outlined the corresponding treatment algorithms. In addition, we developed a step-by-step workup for spinopelvic mobility and total hip arthroplasty (THA). Normal spinopelvic mobility changes from standing to sitting; the hip flexes, and the posterior pelvic tilt increases with a concomitant increase in acetabular anteversion and decreasing lumbar lordosis. Most classifications are based on a division of spinopelvic mobility based on ΔSS (sacral slope) into stiff, normal, and hypermobile, and a categorization of the sagittal spinal balance regarding pelvic incidence (PI) and lumbar lordosis (LL) mismatch (PI–LL = ± 10° balanced versus PI–LL > 10° unbalanced) and corresponding adjustment of the acetabular component implantation. When performing THA, patients with suspected pathologic spinopelvic mobility should be identified by medical history and examination, and a radiological evaluation (a.p. pelvis standing and lateral femur to L1 or C7 (if EOS (EOS imaging, Paris, France) is available), respectively, for standing and sitting radiographs) of spinopelvic parameters should be conducted in order to classify the patient and determine the appropriate treatment strategy. Spine surgeons, before planned spinal fusion in the presence of osteoarthritis of the hip, should consider a hip flexion contracture and inform the patient of an increased risk of complications with existing or planned THA.

## 1. Spinopelvic Mobility

The spine, pelvis, and hips take part in dynamic and complex interaction with one another. Spinopelvic mobility describes the underlying concept, which is necessary for normal human movement and posture [1,2,3,4]. From standing to sitting, adaptation processes are performed; the sacrum moves posteriorly, the lumbar lordosis decreases, and the acetabular anteversion increases [5]. Only part of the movement is performed by the hip joint. More precisely, the hip bends about 55–70°, the pelvis tilts back (posterior APPt (anterior plane pelvic tilt)) approximately 20°, and the lumbar spine lordosis decreases by about 20° [1] (Figure 1). The posterior tilt of the pelvis reduces the sacral slope to the same extent. By bending the lumbar spine, sagittal balance is maintained when changing position [2]. For each degree (1.0°) of posterior pelvic movement, there is an increase of 0.7° to 0.8° in acetabular anteversion [3,4]. When changing position from standing to supine, the pelvis moves anteriorly and leads to a reduction in acetabular anteversion [6]. The anterior tilt of the pelvis is performed to a smaller extent than the posterior tilt. One should be aware that the spinopelvic parameters are dynamic and change in different positions to ensure movement and posture.

Spinopelvic mobility can be affected by degenerative diseases of the spine and hip and by spinal fusion surgery. In an aging society, the prevalence of degenerative musculoskeletal diseases increases. As a result, there will be more patients with concurrent degenerative spine and hip pathologies [7]. Accordingly, the number of patients requiring lumbar fusion and total hip arthroplasty (THA) will increase considerably [8,9,10,11]. To compensate for degenerative changes in the lumbar spine, mechanisms are being developed to maintain sagittal balance and efficient posture [12]. Alterations in spinopelvic mobility also affect the functional orientation of the acetabulum. Therefore, attention was recently focused on the relationship between spinopelvic mobility and implant positioning in THA and the prevention of complications [13]. Two examples illustrate the importance of spinopelvic mobility in THA patients, its changes, and possible consequences; iatrogenically or degeneratively caused stiffness of the spine and the associated loss of posterior tilt can lead to anterior impingement and subsequent posterior dislocation when sitting. While standing, a degenerative kyphotic spine is compensated for by posterior tilt of the pelvis, which may pose a risk for posterior impingement and subsequent anterior dislocation.

Since the topic is of enormous relevance due to the large number of affected patients and due to the fact that the complex interaction of the spine, pelvis, and hip is not yet fully understood, we reviewed spinopelvic mobility based on the current literature. We define the different terms, investigate the necessary imaging, analyze different classifications of spinopelvic mobility, and identify the consequences to be drawn and possible surgical strategies with a particular focus on hip and spine surgeons.

## 2. Common Terms

In order to classify the different terminology related to spinopelvic mobility, we explain the most common terms (Table 1, Figure 2). Unfortunately, similar terms are used by spinal and arthroplasty surgeons with varying definitions. The term “pelvic tilt” is used in arthroplasty in relation to the rotation of the pelvis in the sagittal plane. The (anterior or posterior) pelvic tilt describes here the angle between the anterior pelvic plane (APP) (reference plane for computer-assisted cup implantation in THA, defined between the two anterosuperior iliac spines and the anterior surface of the pubic symphysis) and the coronal plane of the body. Thus, we consider this pelvic tilt as anterior pelvic plane tilt (APPt). APPt can be performed both anteriorly and posteriorly [14]. Posterior APPt describes a backward motion of the pelvis and is the equivalent motion to pelvic retroversion. This might be confusing because in hip arthroplasty the term “retroversion” is used for the acetabular cup. However, posterior APPt (or pelvic retroversion) leads to an acetabular opening and, thus, increased anteversion. Confusingly, the term (spino)pelvic tilt ((s)PT) is used by spinal surgeons as a spinopelvic parameter. It is defined as the angle between a vertical reference line and a line between the center of the S1 endplate and the femoral heads (Figure 3). (Spino)pelvic tilt is also involved in fixed function. The sum of (spino)pelvic tilt and sacral slope (SS) is the pelvic incidence (PI), a position-independent interindividual different anatomical parameter (PI = SS + (s)PT)). When changing position, the change in pelvic tilt (APPt and (s)PT) correlates inversely with the change in sacral slope, and this correlates directly with lumbar lordosis to maintain an upright posture.

## 3. Abnormal Spinopelvic Mobility

Age-related changes in the spine and pelvis can influence spinopelvic mobility. Degenerative spinal changes can lead to a sagittal imbalance. This is caused by increasing kyphosis [18]. In order to maintain a sagittal balance with a painless erect posture, compensatory mechanisms such as an increase in posterior APPt are applied [19]. Compensation by posterior APPt is limited by anatomical conditions such as the hip extension reserve [20]. Increased posterior APPt leads to an enhancement of acetabular anteversion; therefore, standing THA patients are at risk of posterior impingement with anterior dislocation [21,22]. In addition, the mobility and associated compensation mechanisms of the spine must be considered. These mechanisms may be reduced by degenerative processes or by iatrogenic spinal fusion. During the transition from standing to sitting, there is an average posterior APPt of 20° (corresponding to ΔSS 20° (Delta sacral slope)) in a normal spine. As the pelvis moves backward, the acetabulum opens, increasing functional anteversion to allow flexion of the femur in the hip joint [23]. Luthringer and Stefl et al. defined spinal stiffness at ΔSS < 10° and ΔSS < 20°, respectively [24]. Patients with spinal stiffness due to degenerative changes or spinal fusion cannot increase their functional acetabular anteversion when changing position for sitting (fixed anterior APPt), instead trying to compensate with femoral hyperflexion at the risk of anterior impingement and consequent posterior dislocation [25].

## 4. Which Parameters to Measure

In preoperative preparations before THA, a detailed medical history and thorough physical examination form an indispensable basis. Preoperative X-ray evaluation should be considered especially in patients with suspected limited spinopelvic mobility, such as patients with history of lumbosacral fusion, kyphotic standing posture, severe spinal degenerative disease, and hip flexion contractures. For imaging, an a.p (anterior posterior). pelvis X-ray, as well as a lateral X-ray from L1 or C7 (if EOS is available) to the proximal femur including the pelvis in sitting (90° flexion in knee and hip, with femurs parallel to the floor using a height-adjustable chair and “relaxed seated” position) and standing positions, should be performed with either normal flat plate radiographs or stereoradiographs (EOS imaging, Paris, France) [26]. Several efforts were made to evaluate spinopelvic mobility, especially pelvic tilt, from a.p pelvis X-rays [27,28]. Nevertheless, uncertainties remain, and lateral functional images (standing and sitting) are preferable for a comprehensive measurement of spinopelvic mobility. A standing a.p. pelvis X-ray should be favored, since the functional position of the acetabular cup depends on the pelvic tilt (APPt), which can differ considerably between the standing and supine positions. The difference between the coronary plane and the anterior pelvic plane (APP) is the anterior or posterior pelvic tilt (APPt) (Figure 4). It is important to note that, with every 1° of posterior tilt (posterior APPt), patients show an increase in acetabular anteversion of 0.7–0.8° and, thus, functional anteversion changes [29,30,31]. The sacral slope (SS) and APPt can be used to assess spinopelvic mobility in lateral X-rays when changing position from standing to sitting (with ΔSS < 10° between standing and sitting defining spinal stiffness, according to Luthringer et al.) [26]. A more advanced concept is the measurement of PI–LL (pelvic incidence–lumbar lordosis) mismatch. A PI–LL mismatch of >10° in the lateral standing X-ray may indicate a flatback deformity. This spinopelvic abnormality should be considered by arthroplasty surgeons due to the known higher THA dislocation risk in this patient cohort [32]. Other parameters to be measured in the lateral X-ray images in sitting and standing positions include AI (anteinclination), PFA (pelvic femoral angle), SAA (sacroacetabular angle), CSI (combined sagittal index), and the previously mentioned LL and PI. CSI is a newly introduced parameter defined by Heckmann et al. as the sum of the cup anteinclination (AI) and PFA in standing and sitting positions. Following their findings, increased standing CSI and fixed posterior APPt are associated with late anterior THA dislocation, while decreased sitting CSI and femoral hyperflexion are associated with late posterior THA dislocation [33]. Another study showed the large variance in spinopelvic mobility pre- and postoperatively in THA, and preoperative prediction of spinopelvic mobility based on imaging remains a challenge [34].

## 5. Why We Should Take a Closer Look

Following the previous findings, performing THA requires more than considering an a.p. pelvis radiograph. In general, THA is a very successful medical intervention. In rare cases, however, complications such as THA dislocations may occur. Patient-, surgeon-, and implant-specific factors were identified [35]. Nevertheless, there is still a considerable number of THA dislocations without known cause. This led to increased attention paid to abnormal spinopelvic mobility in terms of THA and cup positioning [4,22,24,26,34,35]. The incidence of instability after primary THA is reported to be approximately 1.5% to 4.8% [34,35,36,37,38]. Focusing on patients with suspected abnormal spinopelvic mobility due to degenerative spinal diseases and after spinal fusion surgery, an increased incidence of dislocations in THA of 7.4% to 8.3% was reported [9,39,40,41]. Supporting these findings, the risk of THA dislocation in patients with lumbar fusion before THA was found to be 7.4%, compared to 4.8% in those without fusion [9]. Malkani found hip instability to be the most common reason for failure leading to revision surgery in patients with lumbar fusion before THA [9]. Gausden reported history of spinal fusion as the strongest independent predictor of dislocation in THA [42]. A meta-analysis including six studies revealed a twofold increased risk of dislocation and a threefold increased risk for revision surgery in patients with spinal fusion [40], whereas Perfetti et al. reported spinal fusion patients to be seven times more likely to experience THA dislocation [43]. In addition to the increased risk of dislocation, prior lumbar fusion is also associated with poorer PROMs (patient-reported outcome measures) after THA [40,44]. Buckland and Sing et al. proved the correlation of an increased dislocation rate after THA with a larger number of spinal fusion levels [45,46]. Sagittal spinal deformity increases the risk of hip dislocation in THA [32]. In summary, restricted spinopelvic mobility, regardless of whether it is iatrogenic or degenerative, has an influence on the alignment of the acetabulum and is considered to increase risk for dislocations in THA. Patients who underwent ASD (adult spine deformity) correction demonstrated a postoperative reduction of acetabular anteversion [39], possibly leading to impingement and dislocation. It is worth remembering that spinopelvic mobility progressively decreases. Tamura et al. reported on these changes in a 10-year follow-up after THA, noting an increase of posterior pelvic tilt (posterior APPt, referred to as PSI (pelvic sagittal inclination) in this study) in standing position [47,48]. Decreasing spinopelvic mobility can be a considerable factor, especially in late-occurring dislocations (more than 1 year) after THA, as one study reported that 90% of late THA dislocations involved a spinopelvic imbalance [33]. It has to be borne in mind that spinopelvic imbalance does not necessarily lead to dislocation, as the majority of THA patients remain without complications despite progressive spine degeneration [49].

## 6. Classifications

For arthroplasty surgeons, the challenge is to find a balance between stable component implantation with the lowest possible risk of dislocation and minimal wear. The different entities of changes in spinopelvic mobility were already classified by some authors.

Phan et al. developed four categories that consider sagittal spinal balance (sPT < 25°; PI–LL < 10°) or imbalance (sPT > 25°; PI–LL > 10°) and flexibility or rigidity (Figure 5). Patients in the flexible and balanced groups have no lumbospinal fusions and no pronounced degenerative spinal changes, and they have unrestricted spinopelvic mobility and a neutral sagittal spinal balance. Patients in the rigid and balanced group have limited spinopelvic mobility due to degenerative changes or spinal fusion, with a balanced sagittal spine, but a lack of compensation capabilities when changing position. Flexible and unbalanced (PT > 25°; PI–LL > 10°) patients due to, for example, neuromuscular kyphosis (e.g., Parkinson’s disease) compensate for their sagittal imbalance with an increased posterior pelvic tilt (posterior APPt) while standing, possibly leading to posterior impingement and anterior dislocation during hip extension. Rigid and unbalanced patients show ankylosis or long lumbosacral fusion with an unbalanced spine in sitting and standing positions, without compensatory mechanisms due to rigidity [50].

Kanawade et al. investigated how anteversion and inclination of the cup component is affected when changing position. They defined spinopelvic mobility via the posterior pelvic tilt (posterior APPt) from standing to sitting in three categories. A posterior ΔAPPt of 20–35° was considered normal, <20° as stiff, and >35° as hypermobile [51].

Stefl et al. classified spinopelvic mobility into five groups based on lateral radiographs of the pelvis and spine evaluating the sacral slope (SS) (referred to as sacral tilt (ST)), anteversion and inclination (referred to as anteinclination (AI)), and sacroacetabular angle (SAA) between standing and sitting positions. They distinguished between normal spinopelvic mobility, hypermobile (∆SS > 30°), fused hip (∆SS < 5°), stuck standing (fixed anterior pelvic tilt with no posterior shift while sitting, SS > 30° in standing and sitting), stuck sitting (fixed posterior pelvic tilt, with no anterior pelvic tilt while standing, SS < 30° in standing and sitting), and kyphotic (SS < 5° with undefined mobility). Their study shows that most spinopelvic imbalances can be compensated for by implantation with adjusted anteversion and inclination of the cup. Even hips with severely restricted spinal mobility can achieve acetabular opening by means of large acetabular angles of the components. However, there remains a higher risk of impingement and subsequent dislocation in patients with pathological spinal imbalance that cannot be compensated for even by optimal cup positioning [24].

Luthringer et al [26]. defined a classification based on the presence of spinal deformity (normal PI–LL = ± 10° versus flatback deformity PI–LL > 10°) and spinopelvic mobility (normal ΔSS > 10° from standing to sitting versus stiff ΔSS < 10°) into four groups (1a, 1b, 2a, 2b). Groups 1a and b are defined as patients with normal sagittal alignment; Group 1a shows normal spinopelvic mobility, while Group 1b is defined as limited spinopelvic mobility with ΔSS < 10° from standing to sitting. Groups 2a and b are defined as patients with flatback deformity (PI–LL > 10°); Group 2a shows normal spinopelvic mobility, while Group 2b is defined as limited spinopelvic mobility with ΔSS < 10° from standing to sitting. Patients with flatback deformity show an increased posterior pelvic tilt (posterior APPt) with correspondingly enlarged functional cup anteversion (comparable to “stuck sitting” by Stefl et al. [24] or the imbalance type by Phan et al. [50]) and associated risk of posterior impingement and anterior dislocation while standing. Additionally, in Group 2b, the challenge is due to the limited spinopelvic mobility, aiming for an increase in cup anteversion in order to avoid anterior impingement and posterior dislocation while sitting. This results in a narrow “safety zone” for cup positioning, with the highest risk of THA instability in these patients leading to the consideration of dual-mobility articulation [26].

These categories of spinal imbalance and spinopelvic mobility can provide general guidance. In principle, the needs of each patient should be considered individually. Due to degenerative processes, imbalance and spinopelvic mobility may further deteriorate over time and the classification of patients may, thus, change. Interestingly, THA and the resulting released hip flexion contractures may improve spinopelvic mobility, which is reflected in an increase in ΔSS postoperatively when changing position from standing to sitting [52,53].

## 7. Implications for Hip Surgeons

Arthroplasty surgeons use the long-established “safe zone” (cup anteversion 15 ± 10° and inclination 40 ± 10°) according to Lewinnek (denoted the LSZ) [54]. The static position (supine or lateral during THA) does not take into account the functional changes caused by pelvis rotation when changing positions from standing to sitting. In general, a more functional position including pelvis rotation is considered in standing a.p. pelvis radiographs compared to supine radiographs; therefore, a standing a.p. pelvis reference radiograph is preferred [26,55]. Lazennec was one of the pioneers in investigating the complex interactions during movement between the hip and spine and their influence on THA. Accordingly, the “functional acetabular cup position” introduced by Lazennec should be considered when performing THA [56]. In recent years, increasing concerns were raised about the LSZ. Tezuka et al. found in their study that THA patients within the LSZ, but outside the functional safe zone based on the measurement of CSI, had a probably higher risk of dislocation despite “correct” angles regarding the LSZ [57]. Similar results were obtained in a study investigating THA dislocations, in which the majority (58%) of the acetabular cup was placed in the LSZ [58]. Murphy et al. recommend pelvic-tilt-adjusted acetabular cup implantation [59]. Following the classifications of spinopelvic mobility introduced above, corresponding treatment regimens are presented. Deviating recommendations of the various authors reflect the not yet fully developed scientific consensus on this topic.

Phan et al. defined four groups based on spinopelvic mobility (flexible or rigid) and sagittal spinal balance (balanced or unbalanced). For the flexible and balanced group, they recommend an acetabular anteversion of 5–25°. The rigid and balanced group presents reduced acetabular anteversion due to a decreased posterior pelvic tilt while sitting. To avoid anterior impingement and posterior dislocation in a sitting position, anteversion at the superior end of the safe zone of 15–25° should be aimed for. The flexible and unbalanced group shows increased acetabular anteversion due to increased posterior pelvic tilt during standing as compensation for spinal imbalance. There are two possible treatment pathways for the flexible and unbalanced and rigid and unbalanced spine patients. The first option is performing spinal correction surgery before THA, leading to patients with balanced spine, and then aiming for the corresponding procedure for the rigid and balanced group. The second option, when prior spinal surgery is refused, is performing cup positioning with reduced anteversion to reduce the risk of posterior impingement in kyphotic patients when standing. The second option involves the risk that a THA revision procedure may be necessary if the patients undergo a surgical spinal realignment [50].

Kanawade et al. defined three groups based on spinopelvic mobility. For the patients with stiff pelvis (ΔAPPt < 20°), they recommend increased anteversion and inclination, and, for the patients with hypermobile pelvis (ΔAPPt > 35°), they recommend reduced anteversion and inclination, compared to patients with normal spinopelvic mobility (cup implantation anteversion 20° and inclination 40°) [51].

Stefl et al. [24] classified patients according to spinopelvic mobility into normal, hypermobile, stiff “stuck standing”, stiff “stuck sitting”, kyphotic, and fused. For hips with normal mobility, they suggest 40° inclination and 20° anteversion with a combined anteversion of 25° to 40°. According to Kanawade et al. [51], a reduction of anteversion (35° to 40°) and inclination (15° to 20°) is recommended for hypermobile and hypermobile kyphotic hips to prevent the cup position being too vertical while sitting. For hips with limited mobility (stiff, “stuck standing”, or “stuck sitting”), they recommend increasing the inclination (45°) and anteversion (20° to 25°), as well as combined anteversion (35° to 40°), to achieve a functional opening of the acetabulum while sitting. Patients with a kyphotic lumbar spine (SS < 5°) and normal mobility may receive standard anteversion and inclination. Special attention should be paid to patients with kyphotic lumbar spine and stiff pelvis. These patients were identified to have a high risk of dislocation; therefore, the implantation of a dual-mobility articulation should be considered [24]. Hip surgeons are not only faced with the challenge of ensuring high implant stability; they also need to consider the range of cup orientations with regards to minimizing wear [55].

Luthringer et al. [26] classified patients into four groups depending on spinopelvic mobility and spinal alignment. For patients with normal spinal alignment (no PI–LL mismatch) and normal mobility (Group 1a), the classical anteversion of 20–25° is recommended. Group 1b shows limited mobility (<10° SS from standing to sitting) and, thus, a reduced posterior pelvic tilt while sitting. Hence, these patients should be given increased anteversion (30°) of the cup to avoid anterior impingement and subsequent posterior dislocation. In Group 2a (flatback deformity and normal mobility), it should be noted that the spinal deformity leads to increased posterior pelvic tilt in a standing position and, thus, to increased acetabular anteversion. Reference should be made to the a.p. pelvis standing image with an anteversion of 25–30°. In Group 2b (flatback deformity and stiff spine), there is a narrow safe zone of recommended increased anteversion (30°) to prevent anterior impingement while sitting without creating too much anteversion while standing, which carries the risk of posterior impingement. Thus, the authors recommend here, similar to Stefl et al., that the use of dual-mobility articulation be considered [24,26].

## 8. Implications for Spine Surgeons

Basically, the question arises in the presence of concurrent spine and hip pathologies as to in which order therapy algorithms can be recommended. Patients with adult spine deformity (ASD) (with a positive sagittal balance due to the loss of lumbar lordosis and PI–LL mismatch) try to maintain an upright posture via mechanical adjustments, namely, hip extension and an increase of posterior pelvic tilt. However, spinal deformity correction surgery aims to change the pelvic tilt (and, thus, the acetabular anteversion); hence, it is suspected that this will have an impact on existing or planned THA [60]. There are a few studies that investigated the change in spinopelvic parameters with pre-existing THA after spinal correction surgery. Postoperatively, they found a reduction in acetabular anteversion and posterior pelvic tilt [61,62]. Patients with spinopelvic malalignment show an increased incidence of abnormally large anteverted acetabular components. After spinal realignment, acetabular anteversion decreased, but this presents patients with a new risk with regard to THA stability [61].

Certain studies investigated the influence of adult spine deformity on THA and found an increased risk of THA dislocation and revision surgery when compared to patients without severe spinal pathologies [32,41,63,64]. Several studies examined the outcome of THA on previous lumbosacral fusion and noted an increased incidence of THA dislocations and revision surgery, with a positive correlation to the number of segments fused (3–7 segments versus one or two segments) [43,45,46,65,66].

Hu et al. showed that acetabular anteversion in patients with ankylosing spondylitis can be restored by pedicle subtraction osteotomy and, thus, subsequent THA can be performed with a lower risk of dislocation [67]. Accordingly, Zheng et al. [68] examined a cohort of patients with ankylosing spondylitis who received both spinal osteotomy and THA ((THA first (*n* = 22)), spinal osteotomy first (*n* = 6)). Only the THA-first group showed THA dislocations immediately after spinal osteotomy. Therefore, the authors recommend spinal osteotomy first, despite the small number of patients examined [68]. It should be noted that both patients with spinal imbalance who received THA and patients who received THA after spinal deformity surgery show increased risk of complications. Nevertheless, patients with advanced osteoarthritis and a spinal deformity requiring surgery should be carefully examined for a hip flexion contracture, as this can promote spinal imbalance but is often improved by THA [69,70,71].

Sultan et al. [39] proposed that patients with hip flexion contractures should first undergo THA and subsequently be re-evaluated regarding the global spinal balance, considering ASD correction for an unbalanced spine. If there is no hip flexion contracture, the more symptomatic pathology should be treated first. If ASD correction is performed first, an anteversion at the superior end of the safe zone should be attempted in the THA that may follow as suggested by Phan et al. [50] (rigid and balanced group). If the THA is performed first, the patient should be advised of the increased risk of complications with spinal imbalance progression and monitoring of spine deformity is recommended [39].

## 9. Discussion

Spinopelvic movement is potentially a complex issue and was long underestimated by both spine and hip arthroplasty surgeons. However, it is relevant due to the predicted increase in the incidence of patients with osteoarthritis and spinal pathologies and the subsequent increase in THA and spinal correction surgery [8,11]. Even though THA is one of the most successful interventions in surgery, complications still occur [72]. Among them, THA dislocation is one of the most severe [35]. Understanding the reasons for THA dislocations is an important step in their prevention. One of the reasons for dislocations is restriction of spinopelvic mobility or spinal imbalance, especially in late THA dislocations [33]. Consequently, arthroplasty surgeons should assess spinopelvic mobility and sagittal balance in their patients before THA. Preoperative X-ray evaluation should be considered, especially in patients with suspected limited spinopelvic mobility, such as history of lumbosacral fusion, kyphotic standing posture, severe spinal degenerative disease, hip flexion contractures, and history of THA dislocation and revision surgery. Since it causes additional strains for patients and surgeons, comprehensive diagnostics should be contemplated at least for high-risk patients. Arthroplasty surgeons often align the implant positioning with the static position in an a.p. pelvis supine radiograph. In order to reflect the dynamic interactions, the functional position of the pelvis should instead be considered using an a.p. pelvis standing radiograph.

When evaluating THA candidates, spinal stiffness (ΔSS < 10° [26] or ΔSS < 20° [51]) should be assessed by performing lateral X-rays of the spine from L1 or C7, including the pelvis and proximal femur, in sitting and standing positions. Stiffness leads to limited posterior pelvic tilt (APPt) from standing to sitting with reduced acetabular anteversion and possible anterior impingement. Therefore, an increased anteversion cup positioning is recommended. The spinal sagittal balance should be examined in an a.p. lateral standing X-ray. In order to provide an adequate statement about the sagittal spinal balance, a spinal radiograph including C7 for the measurement of the sagittal vertical axis (C7-SVA) should be considered. A kyphotic spine with spinal imbalance (PI–LL mismatch > 10°) while standing is compensated for by increasing the posterior pelvic tilt (posterior APPt). This leads to increased acetabular anteversion in a standing position and possible posterior impingement.

Despite efforts to reduce the rate of dislocation by taking into account the spinopelvic parameters, it should be remembered that THA dislocations may have many causes such as surgical approach, soft tissue damage, previous surgery, and malposition of the implants. It should also be mentioned that acute postoperative dislocations are usually not due to abnormal spinopelvic mobility [73].

As recommendations for the evaluation of spinopelvic parameters in THA, we identified these steps as crucial and summarized them as follows:Medical history and precise physical examination (including hip flexion contracture)If spinal and pelvic pathologies are present, identifying leading symptomsIdentifying patients at risk for spinopelvic stiffness and sagittal spinal imbalancePerforming a.p. pelvis standing and lateral standing and sitting radiographs from L1 or C7 (if EOS is available), respectively, to proximal femurClassifying the patients according to spinopelvic mobility and sagittal spinal balanceModifying acetabular component positioning in accordance with the classification and, if necessary, using dual-mobility articulation in high-risk patients.

## 10. Conclusions

Spinopelvic mobility is a substantial factor to be assessed when performing THA. Abnormal spinopelvic mobility is a risk factor for THA dislocation. This is relevant not only for arthroplasty surgeons, but also for spine surgeons, as the coincidence of osteoarthritis and spinal pathologies will increase. The disease patterns influence each other through the complex interplay of hip, pelvis, and spine; therefore, these pathologies should not be considered in isolation. The patients at risk should be identified via their medical history and examination, and appropriate preoperative imaging should be performed (a.p. pelvis standing, lateral sitting, and standing spine radiographs of femur to L1 or C7, respectively). Patients are classified according to their spinopelvic mobility and sagittal spinal balance, and the implant positioning or implant selection (use of dual-mobility articulation) should be adjusted accordingly. Implant positioning should be based more on functional rather than static aspects, and the classic coronal Lewinnek safe zone should at least be critically reviewed.

## Figures and Tables

**Figure 1 jcm-09-02569-f001:**
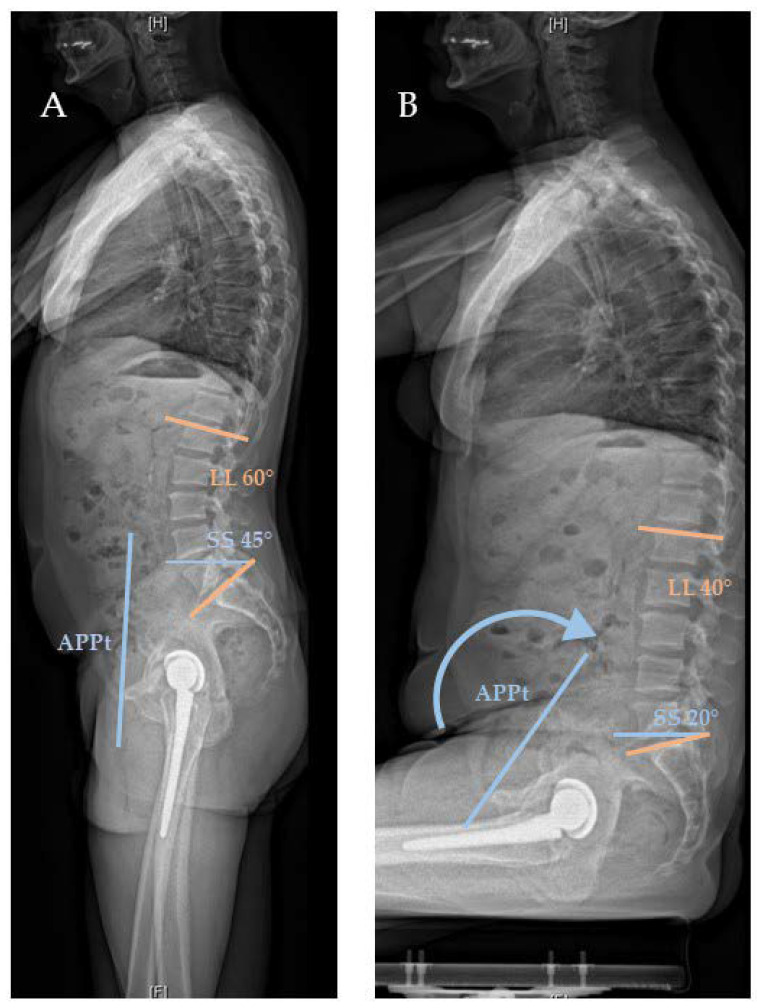
Normal spinopelvic motion with an increase in posterior pelvic tilt (APPt) (blue arrow) and decrease of sacral slope (SS) (blue and orange) and lumbar lordosis (LL) (orange) in lateral standing (**A**) and sitting (**B**) EOS radiographs.

**Figure 2 jcm-09-02569-f002:**
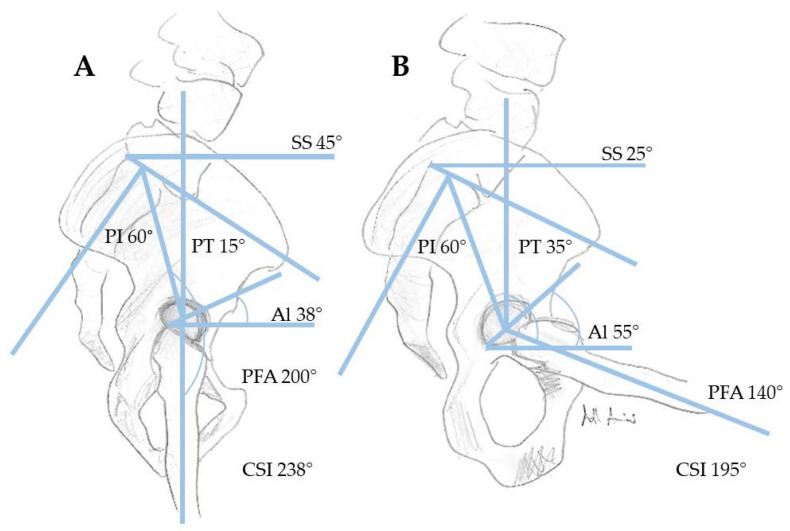
Overview of relevant spinopelvic parameters in standing (**A**) and sitting (**B**) positions with normal spinopelvic motion ((SS, PI (pelvic incidence), AI (anteinclination), PFA (pelvic femoral angle), SS (sacral slope), CSI (combined sagittal index), PT PT (pelvic tilt) (sPT)).

**Figure 3 jcm-09-02569-f003:**
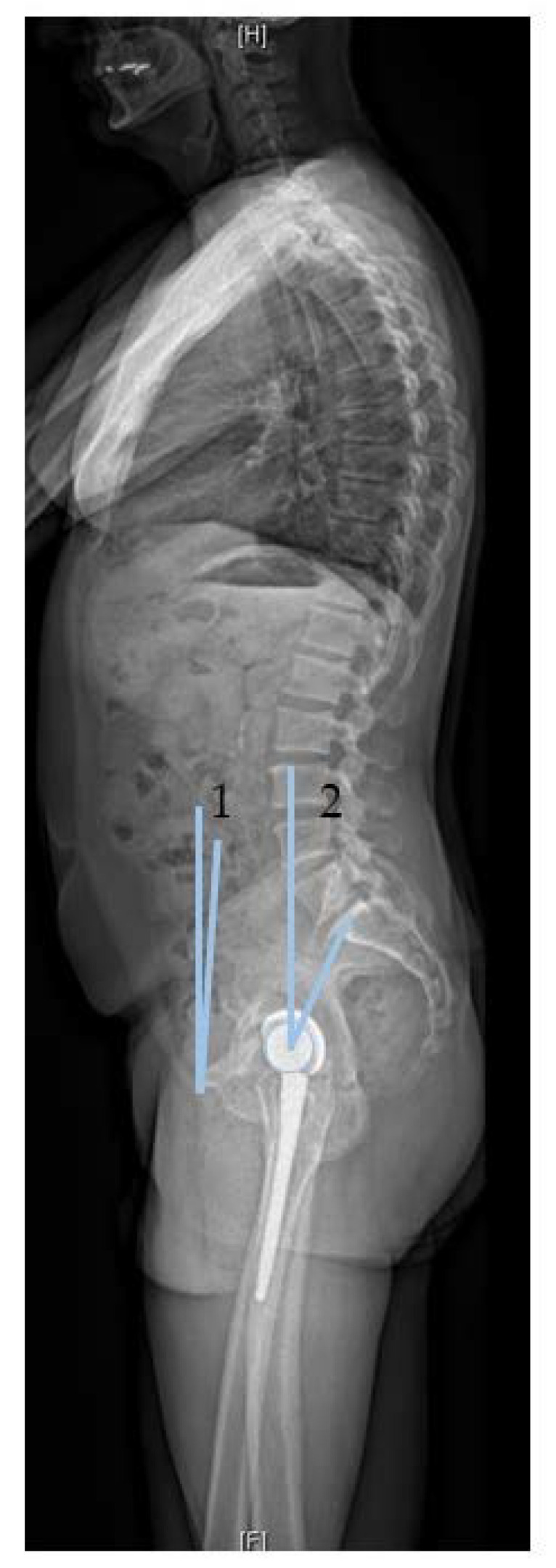
Measurement of the two methods for pelvic tilt: **Method 1**, used by arthroplasty surgeons, defined by the angle between the vertical axis and a line from the femoral heads to the midpoint of the S1 endplate; **Method 2**, used by spine surgeons, defined by the angle formed between the coronal plane and a line from the anterior superior iliac spine to pubic symphysis.

**Figure 4 jcm-09-02569-f004:**
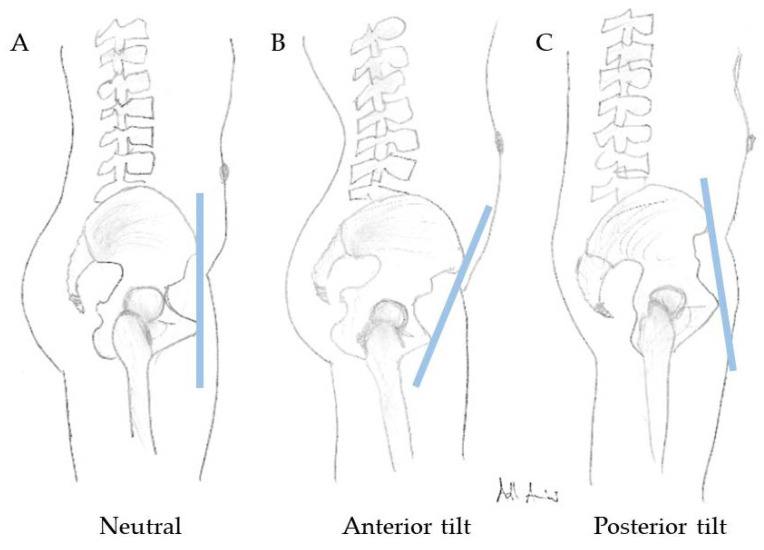
Neutral (**A**), anterior (**B**), and posterior (**C**) pelvic tilt (APPt).

**Figure 5 jcm-09-02569-f005:**
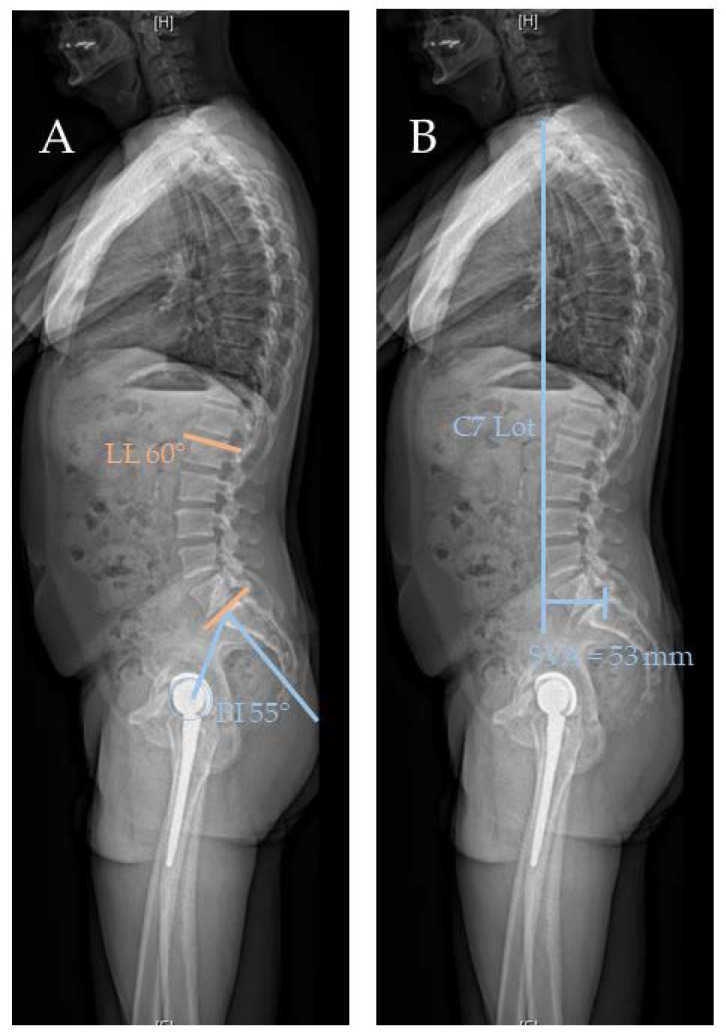
(**A**) Lateral standing radiograph with PI and LL used in PI and LL mismatch (PI–LL > 10°). (**B**) Lateral standing radiograph with C7 plumb line and sagittal vertical axis (SVA). This is an abnormal sagittal balance with an SVA larger than 20 mm.

**Table 1 jcm-09-02569-t001:** Overview of terms related to spinopelvic mobility according to Ike et al., Attenello et al., and Lum et al. [15,16,17].

Term	Definition
Sacral Slope (Tilt) (SS)	Angle Between A Horizontal Line and The Sacral Endplate.
Pelvic Incidence (PI)	Angle Centered at Midpoint of The S1 Endplate, Perpendicular to The Sacral Base and Center of The Femoral Heads. Intraindividual Constant Value That Does Not Change with Sitting and Standing.
(Spino)Pelvic Tilt ((s)PT)	Angle Between the Vertical Axis and A Line from The Center of The Femoral Heads to The Midpoint of The S1 Endplate. Term for Pelvic Tilt Used in Spine Literature Referring to The Position of The Sacrum Relative to The Femoral Heads.
Anterior Plane Pelvic Tilt (APPt)	Term for Pelvic Tilt Used in Arthroplasty Literature Referring to The Rotation of The Pelvis in The Sagittal Plane Defined by The Angle Between the Two Anterosuperior Iliac Spines and The Anterior Surface of The Pubic Symphysis and The Coronal Vertical Plane.
Pelvic Retroversion	Posterior Rotation of The Pelvic in The Sagittal Plane, Equivalent to Posterior Anterior Plane Pelvic Tilt.
Anteinclination (AI)	Angle Between A Line from Anterior to Posterior Acetabular Wall and a Horizontal Reference Line. Changing with Pelvis Motion. The Angle Is Affected by a Combination of Anteversion And Inclination of The Cup.
Sacroacetabular Angle (SAA)	Angle Between Sacral Slope (SS) And A Line from Anterior to Posterior Wall (AI).
Proximal Femoral Angle (PFA)	Angle from The Center of Femoral Heads, Between Midpoint of The S1 Endplate and The Anatomical Femoral Stem.
Combined Sagittal Index (CSI)	Angle of The Acetabular Cup in The Sagittal Plane That is the Sum of The Anteinclination (AI) And the Pelvic Femoral Angle (PFA).
Spinal Imbalance	Abnormal Spinal Mobility Defined by Pelvic Incidence (PI) And Lumbar Lordosis (LL) Mismatch (PI–LL) > 10° Or C7–SVA (Sagittal Vertical Axis)
Hypermobility	Spinopelvic Motion When Transitioning from Standing to Sitting. Defined as >30° ΔSS
Stiffness	Spinopelvic Motion When Transitioning from Standing to Sitting. Defined as <10° ΔSS

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
