# Peer review of "The Impact of Spinopelvic Mobility on Arthroplasty: Implications for Hip and Spine Surgeons"

_jcm, 2020, doi:10.3390/jcm9082569_

Round 1

Reviewer 1 Report

The topic in this article is very important.

The paper is well-written and well-summarized.

I just share two more important articles related to this topic.

Homma Y, Ishii S, Yanagisawa N, et al. Pelvic mobility before and after total hip arthroplasty [published online ahead of print, 2020 Jul 4]. Int Orthop. 2020;10.1007/s00264-020-04688-6. doi:10.1007/s00264-020-04688-6

Ochi H, Baba T, Homma Y, Matsumoto M, Nojiri H, Kaneko K. Importance of the spinopelvic factors on the pelvic inclination from standing to sitting before total hip arthroplasty. Eur Spine J. 2016;25(11):3699-3706. doi:10.1007/s00586-015-4217-2

If those papers are attractive for the authors, please add them.

Thank you very much for a chance to review this important paper.

Author Response

Point 1:

I just share two more important articles related to this topic.

Homma Y, Ishii S, Yanagisawa N, et al. Pelvic mobility before and after total hip arthroplasty [published online ahead of print, 2020 Jul 4]. Int Orthop. 2020;10.1007/s00264-020-04688-6. doi:10.1007/s00264-020-04688-6

Ochi H, Baba T, Homma Y, Matsumoto M, Nojiri H, Kaneko K. Importance of the spinopelvic factors on the pelvic inclination from standing to sitting before total hip arthroplasty. Eur Spine J. 2016;25(11):3699-3706. doi:10.1007/s00586-015-4217-2

If those papers are attractive for the authors, please add them.

Response Point 1:

Dear Reviewer, we highly appreciate the time and effort associated with our manuscript. Following your recommendation we have integrated the two related publications into our work. (Line 35 and 54). If there are any further changes needed from your side, we are willing to address them anytime.

We would like to assure you that we did the English correction offered on MDPI.

Reviewer 2 Report

Thank you for the opportunity to review this piece of work which nicely presents this complex topic.

I think the authors have done a nice job at it ,However, I have specific comments/requests if I may; those include the following:

  1. The comment about obtaining radiographic information from femur to C7 (abstract & text), makes sense however no data presented or studied reports on this. In fact, most of the work is regarding the interaction about femur-hip-pelvis-lumbar spine). Thus, the wording should be changed. C7-Femur can only be studied well enough with EOS, which is not readily available and thus such recommendation should be used with caution.
  2. Lines 48-49 should be rephrased; ‘ To compensate.. efficient posture’ the reference does not report on this. It is not limited to spinopelvic mobility.
  3. Table 1: Acetabular opening is not a commonly used term or measured and I would remove this. It can be used in text as ‘common language’ but not in the Table when all other established parameters measured are described
  4. What type of assessments should be performed quasit-statically? What type of seated measurement/assessment should be performed? Deep-seated or ‘relaxed-seated’? What is the benefit (pros/cons) of each?
  5. Implications for hip surgeons (Section 7). This section is not clear at present; each paragraph presents different data/ recommendation by different papers. In two papers the authors are the same and hence recommendations are slightly different (Kanawade and Stefl). A table should be better and this section should be reduced in text
  6. A section on data about hip- or spine- first in patients with hip-spine syndrome should be discussed. This is important information and there are a few studies out on the topic. This would be as relevant as the information presented in section in Sections 7&8.

I congratulate the authors on this work and I look forward to seeing the improved manuscript.

Author Response

Point 1:

The comment about obtaining radiographic information from femur to C7 (abstract & text), makes sense however no data presented or studied reports on this. In fact, most of the work is regarding the interaction about femur-hip-pelvis-lumbar spine). Thus, the wording should be changed. C7-Femur can only be studied well enough with EOS, which is not readily available and thus such recommendation should be used with caution.

Response Point 1:

First of all we highly appreciate the constructive feedback and valuable comments and your effort to improve our manuscript. We are delighted by the expertise of the reviewer and thankful for her/his insights.

Thank you very much for the important and correct annotation. We recently carried out a prospective study on this subject with EOS at our institution. Nevertheless it is absolutely correct that a general recommendation cannot be based on a diagnostic tool which is not clinical standard.We will adapt this accordingly to “femur to L1 or C7 (if EOS is available), respectively” Please excuse this impreciseness.

Please see line: 22, 120/121, 361, 381/382, 394

Point 2:

Lines 48-49 should be rephrased; ‘ To compensate.. efficient posture’ the reference does not report on this. It is not limited to spinopelvic mobility.

Response Point 2:

We really appreciate the important note and the attentive review. We have adapted the passage accordingly to:" To compensate for degenerative changes in the lumbar spine, mechanisms are being developed to maintain sagittal balance and an efficient posture.”

Point 3

Table 1: Acetabular opening is not a commonly used term or measured and I would remove this. It can be used in text as ‘common language’ but not in the Table when all other established parameters measured are described.

Response Point 3:

Thank you very much for your remark. We have removed "acetabular opening" according to your reasonable suggestion. (please see table 1)

Point 4:

What type of assessments should be performed quasit-statically? What type of seated measurement/assessment should be performed? Deep-seated or ‘relaxed-seated’? What is the benefit (pros/cons) of each?

Response Point 4:

Thank you for the valuable comment. In order to obtain comparable measured values, we recommend a sitting position that is as standardized as possible. On the basis of two  prospective studies, we recommend a sitting posture with approximately 90° flexion in the hips and knees, with the femora aligned parallel to the floor.(1,2) A deep-seated or maximum forward-flexed sitting position in the sense of a functional image can be helpful in determining the anatomical-functional limits of the patient to detect possible impingement situations (and subsequent dislocation).(3) However, it cannot be standardized due to the different abilities of the patient and for example bears the risk of actual THA dislocations. Therefore, we recommend some degree of standardization using a height-adjustable chair with the above-mentioned alignment of the femora parallel to the floor and an otherwise "relaxed-seated" position to achieve the most "natural" impression of the patients posture. We have included this information in the manuscript (please see line 122) and would like to thank the reviewer for pointing this out.

  1. Innmann MM, Merle C, Gotterbarm T, Ewerbeck V, Beaulé PE, Grammatopoulos G. Can spinopelvic mobility be predicted in patients awaiting total hip arthroplasty? A prospective, diagnostic study of patients with end-stage hip osteoarthritis. Bone Joint J. 2019;101-B(8):902-909. doi:10.1302/0301-620X.101B8.BJJ-2019-0106.R1
  2. Esposito CI, Miller TT, Kim HJ, et al. Does Degenerative Lumbar Spine Disease Influence Femoroacetabular Flexion in Patients Undergoing Total Hip Arthroplasty?. Clin Orthop Relat Res. 2016;474(8):1788-1797. doi:10.1007/s11999-016-4787-2
  3. Luthringer TA, Vigdorchik JM. A Preoperative Workup of a "Hip-Spine" Total Hip Arthroplasty Patient: A Simplified Approach to a Complex Problem. J Arthroplasty. 2019;34(7S):S57-S70. doi:10.1016/j.arth.2019.01.012

Point 5:

Implications for hip surgeons (Section 7). This section is not clear at present; each paragraph presents different data/ recommendation by different papers. In two papers the authors are the same and hence recommendations are slightly different (Kanawade and Stefl). A table should be better and this section should be reduced in text.

Response Point 5:

We deeply appreciate your efforts to obtain a most comprehensive study. We also strive for the clearest possible statements and recommendations. In this section we have chosen this structure for two reasons. To provide largest possible clarity we intended to  continue with the structure introduced in the “Classification” section referring to the different authors and their recommendations. The partially different recommendations  of the various authors reflect the not yet fully developed scientific consensus on this topic. Nevertheless, we have attempted to implement the reviewer's recommendation and have shortened the section(please see line 251-253, 255-256, 287-288) and provided the following supplement to the non-uniform recommendations: " Deviating recommendations of the various authors reflect the not yet fully developed scientific consensus on this topic."(line 258-259)

Point 6:

A section on data about hip- or spine- first in patients with hip-spine syndrome should be discussed. This is important information and there are a few studies out on the topic. This would be as relevant as the information presented in section in Sections 7&8.

Response Point 6:

We are very pleased that you have raised this important issue. As there is very likely to be an increase in the simultaneous symptomatic occurrence of OA and degenerative changes in the (lumbar) spine. Since spinal balance or spinopelvic mobility is closely linked to spinal pathologies, this also concerns our review article. The difficulty of clearly recommending which pathology should be surgically addressed first is further increased by the fact that the two pathologies can influence each other. The complexity and controversy of the topic is also reflected in a survey of the leading orthopaedic surgeons (arthroplasty and spine) in the USA.(1) A review article on the hip-spine syndrome recommends a detailed history and physical examination, as well as broad diagnostics before surgical therapy, but the symptom-guided shared decision making between surgeon and patient remains an important measure to find the appropriate therapy (sequence). In section 8 we have tried to approach this complex topic with a focus on spinopelvic mobility and spinal balance and give recommendations for the preoperative management of such patients. We hope we could demonstrate the great relevance of the topic for the authors, but also showing that there are limitations to our review article.  Once again, we would like to thank you.

  1. Liu N, Goodman SB, Lachiewicz PF, Wood KB. Hip or spine surgery first?: a survey of treatment order for patients with concurrent degenerative hip and spinal disorders. Bone Joint J. 2019;101-B(6_Supple_B):37-44. doi:10.1302/0301-620X.101B6.BJJ-2018-1073.R1
  2. L Devin CJ, McCullough KA, Morris BJ, Yates AJ, Kang JD. Hip-spine syndrome. J Am Acad Orthop Surg. 2012;20(7):434-442. doi:10.5435/JAAOS-20-07-434

Reviewer 3 Report

Well reviewed and written paper about spinopelvic mobility and THA.

For better understanding to readers, if you can make and show algorithmic diagram between spinopelvic mobility, spinal deformity and recommended cup orientation, it will be much more helpful in summarizing the concept.

I appreciate your time and effort to make this article in clear fashion.

Author Response

Point 1:

For better understanding to readers, if you can make and show algorithmic diagram between spinopelvic mobility, spinal deformity and recommended cup orientation, it will be much more helpful in summarizing the concept.

Response Point 1:

We greatly appreciate your valuable comments and your efforts to improve our manuscript. We have tried to deal with the complex topics of spinopelvic mobility, spinal deformity/ spinal balance and THA respectively cup positioning in the best possible way. Since there is no final scientific consensus on this topic (spinopelvic mobility/spinal balance and its influence on THA), several classifications with slightly different recommendations exist in parallel. We have tried to reflect this adequately. We consider a diagram suggested by you for a better visual representation of the topic to be very useful, with the restriction that it simplifies and generalizes contexts considerably. It is possible that it does not do sufficient justice to this complex topic. You can find a suggestion of our algorithmic diagram in the appendix. Once again, we would like to thank you.
